# Changes in Cells Associated with Insulin Resistance

**DOI:** 10.3390/ijms25042397

**Published:** 2024-02-18

**Authors:** Leszek Szablewski

**Affiliations:** Chair and Department of General Biology and Parasitology, Medical University of Warsaw, Chałubińskiego Str. 5, 02-004 Warsaw, Poland; leszek.szablewski@wum.edu.pl; Tel.: +48-22-621-26-07

**Keywords:** insulin, insulin cellular signaling, insulin resistance, syndromes of insulin resistance, cellular changes associated with insulin resistance

## Abstract

Insulin is a polypeptide hormone synthesized and secreted by pancreatic β-cells. It plays an important role as a metabolic hormone. Insulin influences the metabolism of glucose, regulating plasma glucose levels and stimulating glucose storage in organs such as the liver, muscles and adipose tissue. It is involved in fat metabolism, increasing the storage of triglycerides and decreasing lipolysis. Ketone body metabolism also depends on insulin action, as insulin reduces ketone body concentrations and influences protein metabolism. It increases nitrogen retention, facilitates the transport of amino acids into cells and increases the synthesis of proteins. Insulin also inhibits protein breakdown and is involved in cellular growth and proliferation. On the other hand, defects in the intracellular signaling pathways of insulin may cause several disturbances in human metabolism, resulting in several chronic diseases. Insulin resistance, also known as impaired insulin sensitivity, is due to the decreased reaction of insulin signaling for glucose levels, seen when glucose use in response to an adequate concentration of insulin is impaired. Insulin resistance may cause, for example, increased plasma insulin levels. That state, called hyperinsulinemia, impairs metabolic processes and is observed in patients with type 2 diabetes mellitus and obesity. Hyperinsulinemia may increase the risk of initiation, progression and metastasis of several cancers and may cause poor cancer outcomes. Insulin resistance is a health problem worldwide; therefore, mechanisms of insulin resistance, causes and types of insulin resistance and strategies against insulin resistance are described in this review. Attention is also paid to factors that are associated with the development of insulin resistance, the main and characteristic symptoms of particular syndromes, plus other aspects of severe insulin resistance. This review mainly focuses on the description and analysis of changes in cells due to insulin resistance.

## 1. Introduction

Insulin is a polypeptide hormone composed of 51 amino acids mainly secreted by pancreatic β-cells as a reaction to an increased concentration of glucose in serum. It is involved in the regulation of different processes in the human body and is considered a metabolic hormone that stimulates several anabolic processes, while catabolic processes are inhibited by this hormone. Insulin regulates energy storage and metabolism in organs such as the liver, kidney, brain, skeletal muscles as well as in adipose tissue. For example, in the liver, it stimulates the synthesis of glycogen and accumulation of lipids and inhibits the production of hepatic glucose in processes of gluconeogenesis and glycogenolysis. In peripheral muscles, insulin stimulates metabolism, increasing glucose uptake, glycogen synthesis and muscle mass. In the brain, this hormone is involved in the stimulation of processes associated with hunger. In adipose tissue, insulin stimulates processes such as metabolism and the uptake of glucose, fat storage in the process of lipogenesis and transport of fatty acids from the bloodstream into cells. Insulin also inhibits lipolysis. There are also several other metabolic processes which are regulated by insulin. By stimulating lipogenesis and the synthesis of proteins and inhibiting lipolysis and protein breakdown, insulin promotes the growth and proliferation of cells. The above-mentioned processes are associated with intracellular signaling pathways [1,2,3,4].

Insulin resistance (IR) is characterized by a reduced response of insulin-sensitive organs and tissues to stimulation by insulin. IR is also described as impaired insulin sensitivity due to a decreased reaction of insulin signaling for blood glucose levels. This pathophysiological reaction increases plasma insulin levels, resulting in hyperinsulinemia, which is associated with an increased risk of initiation, progression and metastasis of several cancers; poor cancer outcomes; as well as the development of metabolic diseases, such as type 2 diabetes mellitus (T2DM), metabolic syndrome, etc. [5]. Genetic and environmental factors contribute to the development of insulin resistance, specifically in mutations in genes that are associated with intracellular insulin signaling pathways. These mutations can alter the insulin protein and/or insulin receptor and can also affect insulin signaling pathways. Other mutations can affect insulin metabolism [4,5,6,7,8].

## 2. Characteristics of Insulin

Insulin controls a wide variety of biological processes. Activation of the insulin receptor (INSR) by hormone binding initiates the signaling pathway that leads to the activation of enzymes which control many aspects of metabolism and growth, which will be further described below. Perturbations in these signaling pathways may cause, for example, insulin resistance, hypertension and/or T2DM [1,5,9,10].

### 2.1. Biosynthesis and Secretion of Insulin

Insulin is an anabolic hormone comprised of 51 amino acids. It has previously been suggested that insulin is solely produced by pancreatic β-cells located in Langerhans islets; however, recent observations have revealed that low concentrations of insulin are also produced in certain neurons of the central nervous system [11]. Insulin’s biosynthesis and secretion are controlled by levels of circulating glucose and increase when glucose levels are high. The biosynthesis of insulin is stimulated when the concentration of glucose is between 2 mM and 4 mM, and secretion is inhibited when glucose levels rise above 5 mM [12,13]. After secretion, the hormone circulates systematically, reaching cells in the liver, muscle and fat, where it is taken and stored, resulting in reduced levels of glucose in the blood [2,14,15].

#### Biosynthesis of Insulin

As mentioned above, the main site of insulin biosynthesis is pancreatic β-cells. In humans, the hormone is encoded by the insulin gene (*INS*), located on the short arm of chromosome 11. The process of transcription is controlled by upstream enhancer elements, such as IDX1 (PDX1), MafA, NeuroD1 and others [2,16]. An effect of translation is synthesized preproinsulin, which, in the rough endoplasmic reticulum (RER), is processed into proinsulin upon cleavage of its signal sequence. Proinsulin is then folded and stabilized in its 3D configuration. This stage is due to linking the semihelical A and B domains by the formation of three disulfide bonds. The next stage of insulin synthesis is processed in the Golgi apparatus, where folded proinsulin is placed in still-immature secretory granules, where the convertases PC1/3 and PC2 cleave the C-peptide. The last process is the synthesis of mature insulin via the removal of C-terminal basic amino acids from the peptide chain. This process is due to carboxypeptidase E [17] (Figure 1). Most insulin granules are stored in the cytoplasm of pancreatic β-cells. To reach the plasma membrane and for secretion of mature insulin by β-cells, granules with insulin must cross a critical actin network. This process requires the reorganization of actin because the actin network plays a role as a physical barrier to the secretion of insulin [2,18,19]. This process is controlled by small G-proteins as well as their activating exchange factors, such as glucose- and Cdc42-dependent activation of Rac 1 [20]. Secretory granules must dock at the plasma membrane. The next step of insulin secretion depends on the response to the intracellular Ca^2+^ signal, which is associated with Ca^2+^ channels [21]. The secretion of insulin is biphasic. The first phase, which is rapid, is caused by fusion and secretion by docked secretory granules. In humans, this phase lasts up to 10 min. The second phase is associated with the reorganization of cytoskeletal elements, namely actin, resulting in the recruitment of granules to the plasma membrane [22].

## 3. Intracellular Insulin Signaling Pathways

Insulin controls many biological processes that involve the tyrosine kinase insulin receptor (INSR). Activated INSR initiates a phosphorylation of substrates associated with insulin signaling pathways, activating enzymes involved in the control of metabolic processes and growth. Disturbances in these signaling pathways may cause insulin resistance.

### 3.1. Insulin Receptor

Insulin action begins when insulin binds to its receptor localized on the cell membrane of the target cells. INSR is a heterotetrameric protein that consists of two extracellular α-subunits and two β-subunits which are bound together by disulfide bonds (Figure 2). Both α- and β-subunits are obtained from a single precursor by proteolytic cleavage. The INSR mRNA undergoes alternative splicing, generating two different isoforms: isoform A (INSR-A) and isoform B (INSR-B). INSR-B contains a 12-amino-acid sequence in the carboxy-terminal (C-terminal) part of α-subunit, which is missing in INSR-A. These isoforms also alter the site of expression. INSR-A is predominantly expressed in embryo and fetal tissues; the central nervous system, especially in the brain; hematopoietic cells; and tumor cells. Expression of INSR-B is highest in the liver and in insulin-targeted tissues, such as muscle, adipose tissue and the kidneys [23,24,25]. INSR-A mediates mainly mitogenic effects, whereas INSR-B is involved mainly in metabolic effects [26]. There is also the hybrid receptor. In this case, the heterotetramer is composed of an α/β dimer of INSR and an α/β dimer of insulin-like growth factor-1 receptor (IGF-1R). Insulin-like growth factor-1 (IGF-1) and insulin-like growth factor-2 (IGF-2) bind with a much higher affinity to the hybrid receptor than insulin [27].

### 3.2. Insulin Receptor Substrates

Insulin binds to the α-subunit of INSR, inducing a conformational change. This conformational change induces activation of tyrosine kinase in the β-subunit, causing the autophosphorylation of tyrosine residues and the subsequent activation of phosphotyrosine-binding proteins [3,4]. The substrates of INSR are, for example, members of the insulin receptor substrate (IRS) family of proteins. These proteins are IRS-1 through IRS-6. IRS-1 and IRS-2 [29,30,31] are expressed principally in human beings. There are also other insulin receptor substrates, such as growth factor receptor-bound protein-2 (GRB-2), GRB-10, Shc-transforming protein (Shc), SH2B adapter protein-2 (SH2B2), protein-tyrosine phosphatase 1B (PTP1B) and others [32]. Phosphorylation of these substrates activates two primary signaling pathways: the phosphoinositide 3-kinase (PI3K)/protein kinase B (AKT) pathway and the mitogen-activated protein kinase (MAPK) pathway. The phosphorylation of Shc activates the Ras-MAPK pathway, whereas the phosphorylation of IRS proteins mostly activates the PI3K/AKT pathway [4,28].

### 3.3. The PI3K/AKT Signaling Pathway

Autophosphorylation of the INSR, due to insulin binding, causes activation of the IRS protein. Activation of PI3K by IRS causes activation of downstream AKT/mTOR-network ((serine/threonine protein kinase, also known as PKB—protein kinase B) mammalian target of rapamycin complex) signaling. This process is due to phosphorylation of phosphatidylinositol 4,5-bisphosphate (PIP_2_) to produce phosphatidylinositol 3,4,5-triphosphate (PIP_3_), which phosphorylates and activates AKT and atypical protein kinase C (aPKC). AKT is involved in many cellular functions. Activated AKT phosphorylates glycogen synthase kinase, causing its deactivation, and inhibits the activity of glycogen synthase and ATP-citrate lyase, resulting in the inhibition of glycogen and fatty acids synthesis. Activated AKT by phosphorylation inactivates the mammalian target of rapamycin complex 1 to promote protein synthesis. By inhibition of the proapoptotic pathway, AKT mediates cell survival. It also activates sterol regulatory binding proteins (SREBPs), which are translocated to the nucleus, where they play a role in the transcription of genes associated with the synthesis of fatty acids and cholesterol. AKT also regulates the cell cycle and translocation of glucose transporters from the intracellular space to the plasma membrane of muscles and fat cells, increasing the uptake of glucose by these cells from the blood [3,6,33,34,35].

### 3.4. The MAPK Signaling Pathway

A second essential insulin signaling pathway is the MAPK signaling pathway. Its activation is independent of the PI3K/AKT signaling pathway [1]. Activated INSR and IRS proteins contain docking sites for adaptor molecules, which contain SH2 domains, such as growth factor receptor-bound protein 2 (Grb2) and Src homology (Shc). The MAPK pathway is activated when IRS-1 binds to Grb2 [4]. To Grb2 binds son-of-sevenless (SOS), which is a guanine nucleotide exchanger factor (GEF) for Ras. SOS binds to Ras, catalyzing the change of Ras from an inactive GDP-bound form (guanosine diphosphate) (Ras-GDP) to an active GTP-bound form (guanosine triphosphate) (Ras GTP). Activated Ras stimulates cRaf that phosphorylates and activates MAPK/Erk kinases, MEK1 and MEK2 (MAP kinases ERK1 and ERK2—extracellular signal-regulated kinase 1 and 2). Then, activated ERKs are translocated to the nucleus, where they cause phosphorylation and transcriptional activation by transcriptional factors, such as ELK1, promoting cell division, protein synthesis and cell growth [3,6,36].

## 4. Role of Insulin in Selected Cells, Tissue, Organs

As mentioned above, the main role of insulin is the regulation of the body’s energy supply. Insulin also contributes to the uptake of glucose from the blood stream into insulin-dependent cells of muscle, the liver and adipose tissue. Insulin may play a specific role in particular cells, depending on the type of cell. These specific roles depend on the intracellular insulin signaling pathway.

### 4.1. Insulin Signaling Pathway and Its Role in the Liver

The primary organ of insulin action is the liver. Insulin action begins after the hormone binds to INSR, which activates IRS-1 [2,37]. Activated IRS-1 recruits PI3K and then AKT. After activation, AKT suppresses the expression of the forkhead box O1 (FOXO1)-mediated gluconeogenic gene, decreasing gluconeogenesis. Insulin increases the synthesis of glycogen in the liver. This is due to the regulation of glycogen synthase 2 (GYS2) and glycogen phosphorylase. Glycogen synthase kinase 3 (GSK3) and protein phosphatase 1 (PP1) are involved in this process. Upregulating sterol regulatory element-binding protein 1c (SREBP-1c), insulin increases de novo lipogenesis in the liver [3,4,38,39].

### 4.2. Insulin Signaling Pathway and Its Role in Muscle

As in previous cells, the intracellular insulin signaling pathway begins when insulin binds to INSR, resulting in the activation of IRS and recruitment of PI3K, activating the AKT signaling pathway. Activated AKT stimulates the uptake of glucose due to the translocation of glucose transporter 4 (GLUT4) storage vesicles from the cytoplasmic milieu to the plasma membrane. The translocation of GLUT4 takes place within minutes of insulin binding to INSR [2]. This translocation depends on inactivation of the GTPase-activating protein (GAP) AKT substrate of 160 kDa (AS160), and the GTP-bound form of Ras-related C3 botulinum toxin substrate 1 (RAC1) must be involved. The stimulation of glycogen synthesis by insulin is associated with the inhibition of glycogen synthase kinase 3, activation of glycogen synthase as well as inactivation of glycogen phosphorylase, which depends on the dephosphorylation of phosphorylase kinase [3,4,40].

### 4.3. Insulin Signaling Pathway and Its Role in White Adipose Tissue

In adipocytes, insulin suppresses lipolysis, which in turn causes the suppression of hepatic glucose production (HGP) [2]. The suppression of HGP is due to the reduction of substrates which are involved in the process of gluconeogenesis. In this process, phosphodiesterase 3B (PDE3B), protein phosphatase 1 and protein phosphatase-2A (PP2A) also play an important role. In adipose tissue/adipocytes, insulin stimulates other processes, such as the transport of glucose into adipocytes, due to the translocation of GLUT4, lipogenesis and adipogenesis [2,41]. In these processes, Rab, SREBP-1c and peroxisome proliferator-activated receptor-γ (PPAR-γ) are involved, respectively. Insulin also activates several substrates, which are associated with processes taking place in adipose tissue cells. Examples of these substrates/enzymes include PIP2, PIP3, PKB, mTORC1 (also known as mechanistic target of rapamycin complex 1), glucose-6-phosphatase (G6Pase), phosphoenolopyruvate carboxykinase 1 (PEPCK; also known as PCK1), glucokinase (GCK), glycerol-3-phosphate acetyltransferase 1 (GPAT1), acetyl-CoA carboxylase (ACC), fatty acid synthase (FAS) and so on [3,42].

### 4.4. Insulin Signaling Pathway and Its Role in Other Organs and Cells

Insulin is also involved in regulating processes in other organs and cells. It regulates the expression and activity of different proteins, such as enzymes, transcription factors, proteins which regulate cell cycle and apoptosis as well as proteins associated with survival [43,44,45].

In the central nervous system, insulin plays an important role. It crosses the blood–brain barrier (BBB) through a receptor-mediated process [42]. In the cerebrospinal fluid, the concentration of insulin is one-third of that in circulation. Insulin action is associated with INSR localized in cell membranes of neurons and glial cells. For example, the mentioned hormone regulates appetite, reducing the expression of neuropeptide Y and agouti-related peptide (orexigenic) and increasing the expression of pro-opiomelanocortin (anorexigenic). It also regulates energy expenditure [46,47,48]. This hormone also participates in the regulation of processes associated with trophic levels and the development of neurons and glial cells. It has also been suggested that insulin contributes to the modulation of cognition, memory and mood [49]. A significantly improved process of memory after intranasal administration of single-dose (160 IU) insulin was observed in women but not in men [50]. Insulin stimulates the uptake of glucose in the spinal cord tissues and brain regions, such as the choroid plexus, the pineal gland and the pituitary [51,52,53]. Results obtained in studies showed that intranasal administration of single-dose insulin decreases food intake in healthy males, although this effect was not observed in women [50]. Precise mechanisms of many insulin actions in the nervous system are still unknown and need further investigation.

Insulin also regulates pancreatic functions. For example, it suppresses the secretion of glucagon from pancreatic α-cells [54], preventing increased glucose levels due to the process of glycogenolysis. It also stimulates δ-cells of pancreatic islets, which secrete somatostatin. As the first target cells for insulin action are the α-cells, these cells located at the periphery of pancreatic islets secrete glucagon. Insulin decreases the secretion of glucagon, which in turn increases many metabolic effects which are associated with insulin action.

In the kidneys, insulin regulates many metabolic processes, such as kidney homeostasis [55] and glucose metabolism [56]. Recently, an association of insulin with metabolic processes was suggested. Therefore, there has been an investigation, for example, into the physiological role of INSR in the kidneys [57,58]. Observations performed on animal models of diabetes and insulin resistance revealed the decreased expression of INSR and phosphorylated INSR in renal epithelial cells. The decreased expression of insulin receptors in the kidneys was detected in rat models of type 1 diabetes and patients with type 2 diabetes [57]. Insulin interacts with angiotensin type 1 receptors in the renin-angiotensin system. Angiotensin II (Ang-II) inhibits the insulin activation of PI3K signaling, causing insulin resistance [59]. Insulin is involved in regulating renal glucose metabolism, as hyperglycemia may lead to the development of diabetic kidney diseases (DKDs) [56]. Insulin is also involved in bone development [60]. This process may be regulated by insulin signaling, by which means insulin promotes osteoblast development and resorption of bones by osteoclasts [57]. It was found that these cells express INSR on their surface [61]. Based on the results, it was suggested that insulin receptors in osteoblasts are necessary for the proliferation, survival and differentiation of osteoblasts. Insulin also increases the rate of osteoblast proliferation, synthesis of collagen, production of alkaline phosphatase and uptake of glucose as well as inhibiting the activity of osteoclasts [62].

Insulin is also involved in metabolic processes of the skin and hair follicles. Hair follicles require a regular supply of oxygen and nutrients [63]. Hyperglycemia decreases oxygen and nutrient supplies, causing follicular damage and changing hair growth. Follicular damage causes thinning, fragility, sparseness of hair and a decline in hair growth [64,65]. Certain skin conditions are associated with insulin action, such as acrochordons and acne [66]. Acne is a symptom of many skin diseases associated with insulin receptors, whereas carbohydrate metabolic impairment is detected in individuals with acrochordons [67]. Deregulated insulin sensitivity is observed in individuals with psoriasis [68,69].

Insulin is also involved in ketone body metabolism. Hyperinsulinemia due to prolonged fasting or uncontrolled diabetes mellitus accelerates fat mobilization, causing an excessive influx of free fatty acids to the liver. This disturbance causes the synthesis of ketone bodies by the liver. Substrates for this synthesis are by-products of incomplete beta-oxidation of long-chain fatty acids. Products of this synthesis, ketoacids such as acetoacetate, beta-hydroxybutyrate and acetone, may be used as a source of energy by extrahepatic tissues, mainly in skeletal muscle and the heart, although under extreme conditions, these derivatives may be used by the brain as a fuel [70]. Concentrations of circulating ketone bodies are reduced by insulin via several mechanisms, for example, the inhibition of lipolysis by hormones and decreased transport of free fatty acids to the liver for ketogenesis. Insulin also inhibits hepatic ketogenesis [71]. Hyperinsulinemia is due to increased peripheral clearance of ketone bodies [72].

Insulin is also involved in protein metabolism. It increases the retention of nitrogen and protein acceleration; increases the number of ribosomes; stimulates the uptake of amino acids by hepatocytes, skeletal muscle and fibroblasts; as well as inhibits protein breakdown. These processes increase protein synthesis and protein concentration [73,74].

Insulin also interacts with the vasculature, influences endothelial cells of the vessels of systemic circulation and acts on endothelial cells, which promotes blood flow and ensures its delivery to peripheral tissues [75]. In large blood vessels, for example, the aorta and the larger arteries, insulin binds to the insulin receptors of endothelial cells. Conformational changes to INSR cause their autophosphorylation, resulting in the phosphorylation of IRS-2. The phosphorylation of IRS-2 activates the PI3K signaling pathway, which signals downstream to the serine and threonine kinase AKT/PKB. AKT activates endothelial NO synthesis, which catalyzes the conversion of L-arginine to NO [76], a potent vasodilator. The stimulation of arteries and arteriole dilation by insulin is due to the synthesis of NO [77]. These vascular effects are temporally associated with insulin action [75]. For example, the full stimulation of glucose uptake by skeletal muscle depends on prior NO-mediated vasodilation [78].

## 5. Insulin Resistance

Physiologically, insulin resistance (IR) is defined “as a state of reduced responsiveness in insulin-targeting tissues to physiological levels of insulin” [3]; “a state of a cell, tissue, or organism in which a greater than normal amount of insulin is required to elicit a quantitatively normal response” [79]; or “an inability of some types of tissues to respond to normal insulin levels, and thus, higher than normal levels of insulin are required to maintain the normal functions of insulin” [3]. The mechanisms of insulin resistance have not been fully established, but several suggestions have been made.

### 5.1. Severe Insulin Resistance Syndromes

Severe insulin resistance syndromes are rare syndromes which are characterized by profound insulin resistance. Severe insulin resistance is defined “as a severely diminished response to insulin’s biological effects and is characterized by substantial hyperinsulinemia and impaired response to endogenous and exogenous insulin” [34]. To maintain euglycemia, patients with severe insulin resistance need large amounts of exogenous insulin. These patients might present hypoglycemia, which may precede hyperglycemia [80,81]. Such syndromes are detected in 0.1–0.5% of patients in diabetes clinics [80].

#### 5.1.1. Pathogenesis of Severe Insulin Resistance

Insulin resistance may be due to different factors, such as mutations in insulin signaling pathways, autoimmune reactions and environmental factors (Table 1).

#### 5.1.2. Defects in the Insulin Gene

Autosomal dominant mutations in the insulin gene influence the secondary structure of the insulin protein due to the disruption of three disulfides and bonds in mature insulin. The impaired secondary structure of insulin causes endoplasmic reticulum stress and destruction of β-cell pancreatic islets of Langerhans. Autosomal recessive mutations are involved in the reduced biosynthesis of the hormone or loss of function of the insulin protein.

Insulin *Wakayama*, a clinical variant insulin, is an effect of the substitution in the A chain of conserved valine by leucine (Val^A3^→Leu). This mutation decreases the affinity of the receptor for insulin binding 500-fold [82]. Mutations in the B chain Phe^B24^→Ser (insulin *Los Angeles*) and Phe^B25^→Leu (insulin *Chicago*) [83] significantly reduce the bioactivity of insulin as well as decrease the binding affinity to the INSR [5]. The substitution of histidine by asparagine (His^B10^→Asp) enhances the activity of the hormone caused by upregulated pathways. This mutation, described in proinsulin, is associated with hyperinsulinemia due to mis-trafficking of the protein [84].

#### 5.1.3. Defects in the Insulin Receptor

The effects of mutations in the insulin receptor gene depend on their site. Mutations in the α-subunit of the insulin receptor may cause a decreased number of active mature insulin receptors as well as the affinity of insulin receptors for binding insulin. The affected activation of downstream signaling cascades, caused by impaired autophosphorylation of the insulin receptor, may be due to mutations in the β-subunit of INSR, the tyrosine kinase domain [25,85].

Donohue syndrome, also known as leprechaunism, is an extremely rare autosomal recessive disease due to mutations in the INSR. This syndrome affects fewer than 1 per million people worldwide, affects the ability of the hormone to bind to the receptor and is the most severe defective insulin signaling syndrome [34]. Patients with leprechaunism seldom live beyond infancy, and most affected individuals survive less than two years. Their deaths are mainly associated with intercurrent infection [80,81].

Rabson–Mendenhall syndrome (RMS) is a rare autosomal recessive disease caused by mutations in the insulin receptor gene, affecting fewer than 1 per million people worldwide. RMS is a mild form of severe insulin resistance syndrome [86]. In affected patients, extremely high levels of insulin [87], fasting hyperglycemia and failed responses to endogenous and exogenous insulin [81] may be detected. Most patients survive only up to 15 years of age, although there have been some cases of affected individuals living into their third decade of life [88].

Type A insulin resistance syndrome (TAIRS) is another rare disorder due to mutations in INSR, with an incidence of 1 in 100 000 [89]. TAIRS may be caused by autosomal dominant or recessive mutations in INSR. Heterozygous and, in some cases, homozygous mutations in the insulin receptor gene impair the function of the receptor as well as signal transduction [90,91,92,93]. TAIRS has a relatively good prognosis, and affected individuals can live beyond middle age [94]. The features of TAIRS are subtler in affected males. In some affected males, low blood sugar (hypoglycemia) can be the only indication. This syndrome is diagnosed more often in females than in males, as females have more health problems which are associated with the condition.

Type C insulin resistance syndrome is a variant of Type A insulin resistance syndrome, with less severe IR. It is also a so-called HAIR-AN syndrome (hyperandrogenic, insulin-resistant, and acanthosis nigricans). This syndrome is inherited as an autosomal dominant disease [95,96]. HAIR-AN syndrome is generally diagnosed in obese women who do not demonstrate defects of the INSR; however, these women may exhibit postreceptor defects [97]. This syndrome may affect up to 3% of women with androgen excess [98]. A significant role in determining the degree of IR may be associated with the degree of obesity [95].

Type B insulin resistance syndrome (TBIRS) is a rare autoimmune disorder. It is caused by polyclonal autoantibodies against the insulin receptor [34]. TBIRS occurs mainly between the fourth and sixth decade of life. Typically, it is diagnosed in middle-aged women in association with other autoimmune disturbances [99,100,101,102,103]. In adulthood, this syndrome is associated with a higher 10-year mortality risk [104]. The autoantibodies may act bi-phasically. In the first phase, they induce hypoglycemia, which ultimately may cause hyperglycemia. It is also suggested that high concentrations of autoantibodies antagonize INSR, causing its inhibition and leading to insulin resistance and hyperglycemia. Low levels of these autoantibodies agonize, causing hypoglycemia [105,106].

#### 5.1.4. Lipodystrophies

The lipodystrophy syndromes are a heterogenous group of disorders. In these syndromes, severe insulin resistance and severe hypertriglyceridemia are observed, leading to pancreatitis and fatty infiltration of the liver, leading to cirrhosis [107]. Complete or partial loss of adipose tissue and depletion of capacity for the storage of lipids [108,109] is detected in affected individuals. Lipodystrophies are rare diseases, with approximately 1.3–4.7 cases per million [110]. The lipodystrophy syndromes have been classified based on the extent, the location of dystrophy and the age of onset [108,109]. According to etiology, there are congenital and acquired lipodystrophies. Based on the extent of adipose tissue deficiency, there are generalized or partial lipodystrophies, in which are included four categories: congenital generalized lipodystrophy; acquired generalized lipodystrophy; congenital partial lipodystrophy; and acquired partial lipodystrophy, also known as Barraquer–Simons syndrome [34]. These lipodystrophies may be inherited in an autosomal recessive manner, for example, congenital generalized lipodystrophy (CGL, Berardinelli–Seip syndrome) and familial partial lipodystrophies 5 and 6 (FPLD5, FPLD6); in an autosomal dominant manner, for example, FPLD1 (Köbberling type), FPLD2 (Dunnigan type), FPLD4, SHORT syndrome (short stature, hyperextensibility of joints, ocular depression, Rieger anomaly, teething delay) and mandibuloacral dysplasia (MAD); and in an X-linked manner, for example, Köbberling–Dunnigan syndrome (however, it rarely may be inherited as autosomal dominant) [34,111]. Lipodystrophies, such as congenital generalized lipodystrophies, are detected in newborns or infants, whereas in the case of acquired generalized lipodystrophy (Lawrence syndrome), patients appear normal at birth, but over days or weeks, they develop lipodystrophy [79].

#### 5.1.5. Other Severe Insulin Resistance Syndromes

Other groups of rare genetic syndromes are described which are associated with severe insulin resistance. These syndromes include Alström syndrome, an autosomal recessive disease; myoclonic dystrophy, an autosomal dominant disorder [79]; autosomal recessive syndromes; Werner syndrome, a progeria and autosomal recessive syndrome [112]; HALS (HIV-associated lipodystrophy syndrome); Bloom syndrome; and microcephalic osteoplastic syndrome [34].

### 5.2. Mechanisms of Insulin Resistance

The mechanism of insulin resistance is not fully understood, although several theories have been suggested. As mentioned earlier, insulin resistance is an important pathology associated with many metabolic diseases, such as T2DM, metabolic syndrome as well as cancers. There is also an alternative theory that insulin resistance is a mechanism that protects tissue from metabolic injury due to nutrient excess [113,114]. Chronic over-nutrition causes those organs and tissues responsive to insulin glucose uptake to protect themselves from toxicity due to nutrients by becoming insulin resistant. This physiological defense may be observed, for example, in the heart and skeletal muscle [115,116]. The glucose-regulating effects of insulin, such as the suppression of hepatic glucose production and lipolysis, uptake of glucose by cells and synthesis of glycogen, in insulin-resistant tissues at normal plasma levels of insulin are not observed [117].

#### 5.2.1. Insulin Resistance in Select Tissues and Organs

Skeletal muscle is an important tissue involved in glucose metabolism in which glucose uptake is stimulated by insulin. This tissue is responsible for up to 80% of postprandial glucose uptake from blood circulation. Therefore, muscular insulin resistance is the primary defect in T2DM and may affect whole-body metabolism [118,119,120]. Insulin resistance in skeletal muscle, due to the deletion of insulin receptor tyrosine kinase (IRTK) or GLUT4, increases adiposity in animal models [121,122]. Results obtained in studies suggest that impaired glucose transport is due to disturbances in the insulin signaling pathway [123]. There are also suggestions that IR in skeletal muscle may be due to defects at the proximal level of insulin signaling. These defects may be caused by impaired activation of insulin receptor tyrosine kinase, IRS-1, PI3K and AKT [3]. This suggestion is confirmed by observations that in skeletal muscle of insulin-resistant mice and obese individuals, the activity of tyrosine kinase of IRTK is decreased [121,124]. It was found also that in insulin-resistant skeletal muscle, the phosphorylation of IRS-1 tyrosine and IRS-1-associated activity of PI3K were damaged [125]. The increased degradation of protein in insulin-resistant muscle due to disturbances in insulin signaling pathways was confirmed also in another study [126]. Experiments performed on animals revealed an increased degradation of protein and activation of the major proteolytic systems, caspase-3 and the proteasome, in the muscle of db/db mice, a model of insulin resistance. More detailed research showed that insulin resistance causes muscle degradation. This pathology is due to a mechanism that suppresses PI3K/AKT, the signaling of which activates caspase-3 and the ubiquitin-proteasome proteolytic pathway, resulting in the degradation of protein [126]. An increased risk of insulin resistance may be due to aging skeletal muscle [127]. With increasing age, the body’s glucose regulation ability decreases and muscles atrophy [128]. Older people have decreased glucose metabolism and expression of GLUT4 in skeletal muscle [129,130], resulting in pathological changes, such as lower activity of AKT stimulated by insulin [131], disturbances in insulin signaling pathways and decreased insulin sensitivity [132]. In aging skeletal muscle, pathophysiological changes are detected, such as mitochondrial dysfunction, intramyocellular lipid accumulation, inflammation, oxidative stress, endoplasmic reticulum stress, autophagy, sarcopenia and a weakened renin-angiotensin system non-classical axis. Aging skeletal muscle is an independent risk factor for insulin resistance, but as mentioned above, pathologies may increase the risk of insulin resistance in skeletal muscle, after which insulin resistance may stimulate developing pathological changes associated with IR [127,133].

The liver is involved in the control of postprandial carbohydrate levels. This control is due to the suppression of hepatic glucose production (HGP) and the stimulation of the deposition of glucose as glycogen. During fasting, the liver plays a role as the primary source of glucose production [134]. Impaired suppression of hepatic gluconeogenesis, a pathology observed in insulin resistance, is mainly caused by defects in lipolysis in adipose tissue and the de-suppression of forkhead box 01 (FOXO1) transcription factor in the liver [134]. Insulin resistance is also associated with impaired stimulation of glycogen synthesis induced by insulin [135,136,137]. Hepatic IR also impairs the regulation of hepatic glycogen metabolism associated with fasting and feeding [136].

Endothelial dysfunction, a cardiovascular disease, is also associated with insulin resistance. As mentioned earlier, insulin stimulates synthesis of the vasodilator nitric oxide. In response to insulin stimulation, synthesized NO increases blood flow to skeletal muscle, increasing glucose uptake [138]. The PI3K-dependent insulin signaling pathway in the endothelium associated with the synthesis of NO is similar to that described in skeletal muscle. The intracellular insulin signaling pathway also regulates the secretion of the vasoconstrictor endothelin-1 (ET-1) in the endothelium [139,140,141]. Insulin resistance impairs the phosphatidylinositol 3-kinase pathway, causing disturbances in the balance between the synthesis of NO and secretion of ET-1, resulting in decreased blood flow and worsened IR. Human and animal studies revealed that endothelial dysfunction due to insulin resistance is associated with oxidative stress, advanced glycation end products (AGE), a hexoamine biosynthetic pathway, proinflammatory signaling, ceramide and inflammation [142].

Insulin resistance may disturb metabolic functions, which is commonly detected in cancer patients. Impaired metabolism causes higher recurrence in rats and decreases the overall survival of these patients [143]. It is suggested that IR may be a primary driver of cancer-associated metabolic dysfunction, which in turn increases the risk of cancer recurrence and risk of death due to cancer [33,143]. A higher risk for the development of cancers such as breast, colorectal, liver, pancreatic, endometrial, lung hepatocellular and prostate is observed in patients with diagnosed insulin resistance [144,145,146]. The progression of cancer is closely related to insulin resistance [147]. The insulin receptor is expressed at higher levels in malignant cells [148]. It is postulated that increased transcription of the genes encoding INSR may be caused by mutations in tumor suppressor genes, such as *TP53*, and the genes which encode BRCA1, von Hippel Lindau (VHL) and Wilms tumor protein (WT1) [149,150]. It was found that in cancer cells, their expression and signaling pathways are often dysregulated [26], and in many cancers, the INSR is overexpressed. Activation of the insulin receptor activates PI3K-AKT-mTOR and MAPK-RAS signaling pathways, which are highly controlled in normal cells, and mutations in these signaling pathways impair activation of these pathways. Results obtained in several studies showed that in solid cancers, mutations in the PI3K and/or RAS are the most common mutations, causing increased signaling through the AKT-mTOR and MAPK, respectively [151,152,153]. It was also found that the MAPK-RAS cascade plays an important role in driving tumor cell proliferation [154].

Insulin resistance in the brain impairs neuroplasticity or the release of neurotransmitters in neurons. The Tau protein participates in the formation of microtubules, which are involved in several cellular processes. The activity of this protein is regulated by phosphorylation caused by insulin. In the brains of patients with Alzheimer’s disease, the level of Tau phosphorylation is three times higher when compared to normal brains. Cognitive dysfunction may be due to brain insulin deficiency and plasma insulin resistance. Insulin resistance influences the development of cognitive dysfunction due to hyperinsulinemia and impaired insulin signaling. Brain insulin resistance that depends on IRS-1 dysfunction may promote cognitive decline, independent of the classic Alzheimer’s disease pathology. For more details, see [155,156,157].

There are also other suggested hypotheses which may explain the mechanisms of insulin resistance.

#### 5.2.2. The Glucose-Fatty Acid Cycle (the Randle Cycle)

Insulin resistance caused by lipids in skeletal muscle, observed as a defect in the uptake of glucose, is due to limited utilization of glucose, stimulated by insulin. Limited utilization of glucose is caused by increased β-oxidation of fatty acids [158,159]. Increased oxidation of fatty acids, mainly in obese subjects, disturbs use of glucose as a main source of energy, inhibiting the activity of glycolytic enzymes. The process of fatty acid β-oxidation may increase mitochondrial acetyl-CoA, causing inactivation of pyruvate dehydrogenase. This action increases levels of citrate and inhibits phosphofructokinase, which is a key glycolytic enzyme. An effect of these biochemical processes is the accumulation of intramyocellular glucose-6-phosphate that inhibits the activity of hexokinase, resulting in the accumulation of intramyocellular glucose. Animal studies performed on rat heart and diaphragm muscles showed that the infusion of fatty acids decreases the utilization of myocellular glucose, increasing intramyocellular concentrations of glucose-6-phosphate [159,160]. Based on the proposed Randle cycle theory, the levels of glucose-6-phosphate and synthesis of glycogen should be increased by fatty acids, whereas glycolysis should be inhibited. Of note, the synthesis of muscle glycogen stimulated by insulin and the oxidation of glucose are suppressed in diabetic patients with chronic insulin resistance [161].

#### 5.2.3. Hexosamine Biosynthesis Pathway

Elevated levels of muscular acetyl-CoA and citrate associated with increased oxidation fat suggests another pathway involved in the regulation of glucose transport. This other mechanism is associated with the hexosamine biosynthesis pathway (HBP) [3]. Fructose-6-phosphate is mainly produced from glucose-6-phosphate. During glycolysis, fructose-6-phosphate is metabolized into fructose-1,6-bisphosphate. Approximately 5% of fructose-6-phosphate is converted to glucosamine-6-phosphate. This reaction is performed by the rate-limiting enzyme of HBP, glutamine:fructose-6-phosphate amidotransferase (GFAT) [162]. Then, glucosamine-6-phosphate is converted to uridine-5′-diphosphate N-acetylglucosamine (UDP-GlcNAc), which plays a role as the donor sugar nucleotide for the glycosylation and O-GlcNAcylation substrates, such as lipids and proteins [162]. Both of these modifications may affect proteins due to the regulation of gene expression or activity of enzymes [163,164,165]. Experiments showed that O-GlcNAcylation modulates insulin sensitivity [166]. O-GlcNAcylation of proteins may compete with phosphorylation for sites involved in regulating the activity of protein and signal transduction [167]. These sites of phosphorylation include insulin signaling pathway components, such as IRS-1/2, PBK, PDK1 and Akt. These components may be modified with O-GlcNAcylation [168,169]. MO-GlcNAc modification of mammalian uncoordinated-18c (Munc18-c), a protein involved in the translocation of GLUT4 stimulated by insulin, was also found. FOXO1, an important transcriptional factor for gluconeogenic genes in the liver, may also be O-GlcNAcylated [170,171].

#### 5.2.4. Ectopic Lipid Accumulation

A single adipose cell has limited capacity for the storage of lipids [172]. A short-term high-fat diet (HFD) causes enlarged adipocytes to stimulate insulin resistance without macrophage infiltration into adipose tissue [173]. Therefore, excess lipids in adipocytes causes insulin resistance without inflammation. This situation may be involved in ectopic lipid accumulation, resulting in IR through the synthesis of metabolically toxic products. In this case, saturated fatty acids increase the production of ceramide, which in turn stimulates insulin resistance [173,174]. The level of diacylglycerol (DAG) in the liver is strongly associated with systemic insulin resistance [172]. Numerous studies have revealed that ectopic lipid accumulation in peripheral tissues, mainly in skeletal muscle and the liver, may cause more severe insulin resistance [175,176,177]. Animal studies showed that lipid accumulation in the liver and skeletal muscle due to HFD feeding or lipid/heparin infusions causes insulin resistance in rats [178]. It was found also that deficiency or inactivation of fat transport proteins, such as CD-36 or fatty acid transport protein 1 (FATP1) stimulates the uptake of glucose in skeletal muscle [179,180]. Liver-specific knockdown FATP2 or FATP5 reduces hepatosteatosis induced by HFD, increasing glucose tolerance [181,182]. It was suggested that ectopic lipid accumulation stimulates insulin resistance and lipid metabolites, such as diacylglycerol, lipophosphatidic acid (LPA), ceramides and acylcarmitines, contributing to the development of insulin resistance in the liver and skeletal muscle.

#### 5.2.5. Diacylglycerol

Insulin resistance induced by lipids, according to the DAG hypothesis, suggests the interference of intracellular insulin signaling associated with the activation of protein kinase C (PKC) due to the accumulation of diacylglycerol within insulin-sensitive tissues [3]. There are three groups in the PKC family: conventional, novel and atypical. Novel PKC (nPKC) has a great affinity for DAG, and results suggest its contribution to insulin resistance [183,184]. Increased levels of DAG in the liver cause the translocation of PKCε, which is the primary hepatic nPKC isoform, to the plasma membrane, inhibiting insulin receptor tyrosine kinase activity. These changes reduce insulin-stimulated activation due to the phosphorylation of IRS-2, PI3K and Akt2 [185,186]. The accumulation of intramyocellular diacylglycerol impairs intracellular insulin signaling pathways and the uptake of glucose by muscle, caused by the activation of PKCθ, a type of nPKC muscle [187,188]. Activated nPKC muscle type causes phosphorylation of IRS-1 stimulated by insulin [189,190]. Performed animal studies confirmed the hypothesis of insulin resistance due to DAG-PKC [183,191].

#### 5.2.6. Ceramide

Ceramide is a sphingolipid. It is an essential bioactive lipid synthesized from an intracellular fatty acid and sphingosine. Ceramide is involved in insulin resistance induced by lipids [192]. It stabilizes cell membranes and regulates the distribution of signaling molecules. Results obtained in research showed a dependence between ceramide levels and insulin resistance. The molecular mechanism that describes the role of ceramide in the induction of insulin resistance has not been clearly demonstrated; there is a hypothesized association of insulin resistance with the impairment of AKT translocation due to the activation of atypical PKCζ and protein phosphatase-2A (PP2A) [193,194]. Ceramide also stimulates insulin resistance by the activation of c-Jun N-terminal kinase (JNK) [195]. Therefore, inhibition of ceramide synthesis causes amelioration of insulin resistance [196]. Unfortunately, there are several unclear and controversial results, meaning that further studies to explain the role of ceramide in insulin resistance are needed [3].

#### 5.2.7. Endoplasmic Reticulum Stress

Obesity enhances endoplasmic reticulum stress (ER); therefore, it was suggested that ER may be associated with hepatic insulin resistance and pancreatic β-cells in obese people. An over-nutrition condition causes the liver to produce an excess of enzymes to process nutrients. After accumulating unfolded proteins in the endoplasmic reticulum, the liver should enhance the unfolded protein response (UPR). In this process, glucose-regulated protein 78 (GPR78, also known as BiP) is involved by the activation of inositol requiring enzyme-1 (IRE1α) and PKR-like ER kinase (PERK). Activating transcription factor 6 (ATF6) inhibits the translation of proteins, promotes ER chaperones and reduces unfolded protein levels [197]. Experiments revealed that induction of endoplasmic reticulum stress suppresses cellular insulin signaling caused by the phosphorylation of serine in IRS-1 by JNK. Increased demand for insulin secretion also induces ER stress in pancreatic β-cells in chronic hyperglycemic conditions [198,199].

#### 5.2.8. Inflammation

Obesity is associated with the development of a chronic low-grade inflammatory state, influencing insulin resistance [200]. Caloric overload causes expansion of adipose tissue, increasing immune cell infiltration, resulting in a proinflammatory response [201]. Proinflammatory cytokines, which induce insulin resistance, may be secreted by adipocytes and macrophages. In adipocytes, increasing secretion of chemokine monocyte chemoattractant protein-1 (MCP-1), a chemokine ligand also known as C-C Motif Chemokine Ligand 2 (CCL2), induces the accumulation of macrophage infiltration into adipose tissue, causing IR [202,203]. Animal studies revealed that deletion of MCP-1 or its receptor (CCR2) improves insulin sensitivity [203]. Immune cells and adipocytes secrete cytokines, such as TNF-α, IL1β and IL-6, the increased secretion of which induces insulin resistance by several mechanisms. Insulin resistance may be due to the activation of Ser/Thr kinases [204,205]; decreased expression of IRS-1, GLUT4 and PPARγ [206]; or activation of suppressor of cytokine signaling 3 (SOCS3) in adipocytes [207]. Toll-like receptors (TLRs) belong to the family of pattern-recognition receptors (PRRs). They play a function in innate immunity and are involved in the identification of tissue injury by danger-associated molecular patterns. During obesity, TLR2 and TLR4 are particularly associated with inflammation-associated insulin resistance. Human and animal studies showed increased expression of TLR4 in adipocytes, hepatocytes, muscle and the hypothalamus due to obesity. Increased levels of TLR4 negatively affect insulin sensitivity [208].

#### 5.2.9. Mitochondrial Dysfunction and ROS Formation

Insulin action is enhanced by low levels of reactive oxygen species (ROS) [209,210]. High levels of ROS may cause oxidative stress. ROS is a by-product of the electron transport chain as a consequence of mitochondrial dysfunction [211]. Changes in mitochondrial proteins due to an increased flux of metabolites into mitochondria and decreased expression of antioxidant enzymes may be due to increased levels of ROS as an effect of obesity [212,213,214,215]. Increased oxidative stress is involved in the activation of stress kinases, which phosphorylate serine in IRS proteins, inducing insulin resistance [216]. Mitochondrial fission, which causes mitochondrial dysfunction, decreases the activation of p38 MAPK, increases the activity of IRS-1 and AKT and causes insulin resistance [217]. Disturbances in mitochondrial functions may increase levels of DAG, causing activation of PKC and decreased phosphorylation of IRS-2 and activity of PI3K [218]. Mitochondrial biogenesis is controlled by nuclear genes, which encode peroxisome proliferator-activated receptor gamma (PPARγ) coactivator 1α (PGC-1α) and PGC-1β. Low expression of these genes decreases the number of mitochondria in muscle, resulting in intramyocellular fat accumulation in patients with insulin resistance [219,220]. Results obtained revealed that increased insulin resistance is associated with decreased PGC-1α beyond normal physical limits [221]. Therefore, activation of PGC-1α may be a therapeutic target for type 2 diabetes mellitus [172].

### 5.3. Selective Insulin Resistance

It was found that in the presence of IR, not all cellular insulin functions are less responsive. In hyperinsulinemia, there are insulin pathways which are highly responsive to insulin. This phenomenon is referred to as selective insulin resistance or pathway-selective insulin responsiveness [222]. For example, in the case of insulin resistance in the liver, insulin not does suppress hepatic glucose production; however, it does stimulate lipogenesis, causing hyperglycemia, hyperlipidemia and hepatic steatosis [222]. The mechanism associated with selective insulin resistance is not fully understood, although there are several suggestions. One hypothesis suggests that selective insulin resistance may be associated with differences in the specificities of substrates involved in AKT phosphorylation in processes of gluconeogenesis and lipogenesis [223]. According to this hypothesis, phosphorylation of AKT Ser473 may activate some AKT substrates that are associated with the process of gluconeogenesis, such as FOXO. Insulin resistance may block these activations. On the other hand, other substrates of AKT, such as glycogen synthase kinase 3 β (GSK3β) and tuberous sclerosis complex 2 (TSC2), the activation of which depends on AKT phosphorylation at Thr308, might be not disrupted, and therefore might be active [224]. Another explanation for selective insulin resistance in the liver suggests the different sensitivities of insulin-induced activation of sterol regulatory element-binding protein 1c (SREBP-1c) and suppression of gluconeogenesis. According to this hypothesis, both processes require specific insulin levels [225,226]. Selective insulin resistance in the liver, according to another hypothesis, may be due to insulin-independent lipogenesis. In the induction of lipogenesis by nutrients, carbohydrate response element-binding protein (ChREBP), the transcription factor peroxisome proliferator-activated receptor-γ (PPARγ) coactivator 1-β and liver X receptor-mediated SREBP-1c [227,228,229] are also involved. Animal studies revealed that these alternative pathways are activated by fructose and monosaccharides [230,231]. For more details, see [3].

## 6. Therapeutic Strategies for Insulin Resistance

Insulin resistance is associated with metabolic disturbances such as T2DM, metabolic syndrome, lipid abnormalities and hepatic derangements. There are suggested treatment strategies against insulin resistance. A sedentary lifestyle and the increased prevalence of obesity contribute to the development of insulin resistance and its consequences. Modifications to lifestyle focused on the reduction of calories and weight as well as increased physical activity are suggested as important factors against insulin resistance [3,232,233]. Suggested proposals include physical activity daily for 30 min and a suitable diet for maintaining a healthy weight, for example, the Mediterranean diet. In the case of severe hypertriglyceridemia, a very-low-fat diet may be appropriate [34]. Exercise can improve skeletal muscle insulin resistance [234]. There are several other strategies in insulin resistance prevention, such as using nature-derived compounds. Note also that there are negative results in the prevention of insulin resistance [235]: patients with diagnosed insulin resistance require insulin therapy. Patients with diagnosed insulin resistance associated with generalized lipodystrophy or mutations in insulin receptors require higher doses of insulin [236,237]. Other drugs may be used, such as insulin-like growth factor-1 in the case of patients with Donohue syndrome [238], insulin sensitizers such as metformin and thiazolidinediones in the case of patients with lipodystrophies [239] and lipid-lowering medications [240]. Etiologic therapeutic strategies, such as leptin therapy, the growth hormone releasing hormone (GHRH) analog and immunosuppressants [34] are also suggested.

## 7. Insulin Resistance Causes Cancer: True or False? Is Insulin Our Friend or Foe?

As mentioned earlier, insulin resistance is involved in the development of different diseases, pathologies and metabolic dysfunctions. IR is associated with hyperglycemia, metabolic syndrome, obesity, hypertriglyceridemia, nonalcoholic fatty liver disease (NAFLD), atherosclerosis, T2DM, cardiovascular disease, PCOS and so on. Insulin resistance impairs insulin secretion from pancreatic β-cells. Disturbances in insulin secretion due to insulin resistance results in compensatory insulin hypersecretion, usually at levels at which normoglycemia cannot be maintained, leading to hyperinsulinemia.

Hyperinsulinemia is a higher than usual amount of insulin in the blood. This pathology is associated with diagnoses of damaged myocardial insulin signaling, mitochondrial dysfunction, endoplasmic reticulum stress, sympathetic nervous system dysfunction, abnormalities in immune response, etc. [6]. Insulin resistance is the main cause of hyperinsulinemia. Hyperinsulinemia may be caused also by a rare tumor of pancreatic β-cells (insulinoma) as well as excessive numbers or growth of these cells (nesidioblastosis) [6]. Hyperinsulinemia is involved in increased morbidity and mortality from cardiovascular complications [241].

Insulin is proposed to be an oncogenic factor [242,243,244] and contributes to the progression of cancer. Therefore, long-lasting hyperinsulinemia due to insulin resistance may promote cancer development, stimulating different intracellular signaling pathways, enhancing cell proliferation associated with enhanced growth factors as well as affecting cell metabolism. Insulin may induce cancer growth by its mitogenic and antiapoptotic effects, as is observed in the case of breast and prostate cancers [33,245,246]. Decreasing levels of sex hormone binding globulin (SHBG) reveals a positive effect on estrogen bioavailability, causing an increased risk of breast cancer [148,247]. Physiological insulin secretion, as is observed in the normoglycemic state, and its action are critical to normal cell growth. In its physiological role in the regulation of cellular growth, insulin interacts with other mediators, such as IGF-1, IGF-2 and their receptors. Epidemiological studies revealed in patients with diagnosed insulin resistance an increased risk for several cancers, such as breast, colorectum, liver and pancreas [145,148]. It is suggested that increased levels of insulin and hyperinsulinemia are associated with the development of a number of cancers, such as colorectal, ovarian, breast and pancreatic cancer [28]. Cross-sectional and prospective studies revealed an association between hyperinsulinemia and an increased risk of breast cancer [248,249]. An association between colorectal cancer and increased serum insulin levels has been found in epidemiological studies [250,251]. Several obtained results revealed that patients with a cancer diagnosis were markedly insulin resistant [143]. In vivo and in vitro studies confirmed suggestions that insulin is involved in the growth of colon epithelial and carcinoma cells [252,253]. Epidemiological studies also revealed a link between elevated levels of insulin and an increased risk of pancreatic cancer. Results obtained in in vitro studies suggest that excessive insulin signaling may be associated with the proliferation and survival of human immortalized pancreatic ductal cells and metastatic pancreatic cancer cells [254,255,256].

As mentioned earlier, isoform B of insulin receptors is up-regulated in cancer [245]. In breast cancer epithelial cells, the insulin receptor is overexpressed up to 10 times the normal amount. This observation may indicate that overexpression of insulin receptors, mainly in the presence of hyperinsulinemia, may be associated with a selective growth advantage to breast cancer cells [28]. Disturbances in intracellular insulin signaling pathways are proposed as a mechanism of breast cancer development. It was found that insulin signaling activates PI3K/AKT/mTOR signaling pathways, which may stimulate proliferation, apoptosis resistance and invasion [257,258,259]. The PI3K/AKT/mTOR signaling pathway is involved in the regulation of central glucose metabolism and aerobic glycolysis. Aggressive cancer cells are glucose dependent, and a large amount of their energy is obtained via aerobic glycolysis, the so-called Warburg effect [260]. Aerobic glycolysis is directly associated with the aggressive biology of cancers. This effect is due to increased glycolysis and uptake of glucose, which supplies anabolic precursors for rapid growth and causes mitochondrial dysfunction, resulting in apoptosis resistance [33]. Based on results obtained, in breast cancer cells, the signaling pathway PI3K/AKT/mTOR is frequently activated [259]. Overexpression of activated AKT predicts poor prognosis in women with breast cancer [261]. There are many studies which have tested the association between the expression of phospho-AKT in breast cancer with overall survival and disease-free survival [261]. All performed studies showed that the activation of AKT signaling is an adverse prognostic factor in breast cancer [33].

Is insulin our friend or foe? The answer to this question was given to us by Paracelsus (1493–1541), Father of Medicine, who wrote: “Omnia sunt venena, nihil est sine venena. Sole dosis facit venenum”: “All substances are poisons; there is none which is not a poison. Only the dose makes the poison”. In the case of insulin, hypoinsulinemia and hyperinsulinemia are “poisons”, whereas normoinsulinemia is not “poison”. Is insulin for us friend or foe? It depends on the dose.

## 8. Conclusions and Perspectives

Insulin resistance, which reduces the responsiveness of cells and tissues to insulin, may be due to both genetic and environmental factors. It is associated with several pathologies and diseases, such as type 2 diabetes mellitus, polycystic ovary syndrome, glucose intolerance, hypertension, metabolic syndrome, cancers, obesity and so on. Severe insulin resistance is involved in clinical syndromes, such as Type A, Type B, Type C, HAIR-AN, Rabson–Mendenhall, leprechaunism and lipodystrophies. The prevalence of obesity and T2DM has increased dramatically worldwide, and individuals with these pathologies are at a higher risk of cancer and other diseases.

Different interactions and distinct molecular, cellular and physiological mechanisms contribute to the relationship between insulin resistance and observed metabolic dysfunctions. Unfortunately, these interactions remain unknown; however, significant progress has been made in understanding the associations between metabolic dysfunctions and severe insulin resistance. Therefore, these associations, mechanisms and interactions need further investigation. Understanding of insulin signaling and insulin resistance will stimulate new therapy designs, including gene therapies and new drugs.

## Figures and Tables

**Figure 1 ijms-25-02397-f001:**
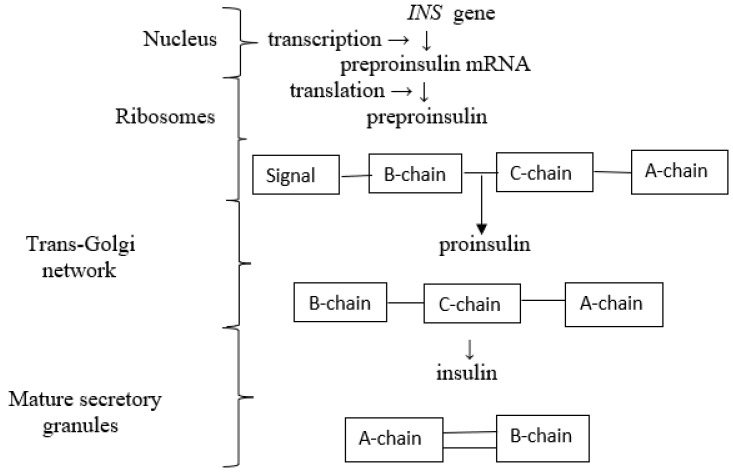
Process of biosynthesis and secretion of insulin. After transcription of preproinsulin mRNA from the *INS* gene, it is translated into preproinsulin peptides. Then, preproinsulin transits through the rough endoplasmic reticulum and trans-Golgi network, causing the preproinsulin to change to its mature form and be stored in mature secretory granules in pancreatic β-cells [2,17,22].

**Figure 2 ijms-25-02397-f002:**
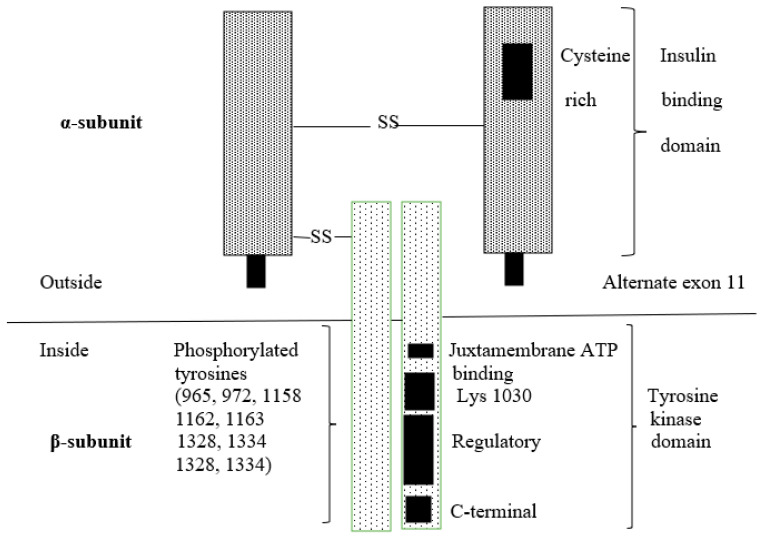
Structure of insulin receptor [4,28]. The insulin receptor is composed of two extracellular insulin-binding α-subunits and two signal transduction β-subunits, which are bound together by disulfide bonds. Conformational change of α-subunits due to binding of insulin to the receptor enables binding of adenosine triphosphate (ATP) to the β-subunit. Binding of ATP activates a tyrosine kinase in the β-subunits, causing autophosphorylation of the receptor. The phosphorylated insulin receptor in turn phosphorylates other protein substrates, such as IRS-1 and IRS-2.

**Table 1 ijms-25-02397-t001:** Characteristics of severe insulin resistance syndromes.

Site of Defects	Cause	Syndrome	Features
Insulin gene	Mutation Val^A3^→Leu	Insulin *Wakayama*	Decreased binding of insulin to the INSR.
	Mutation Phe^B24^→Ser	Insulin *Los Angeles*	Significantly reduced bioactivity of insulin.
	Mutation Phe^B25^→Leu	Insulin Chicago	Decreased binding of insulin to the INSR.
Insulin receptor	Autosomal recessive mutations	Donohue syndrome	Affected patients seldom live beyond infancy. Most affected patients survive fewer than 2 years.
		Rabson–Mendenhall syndrome	Hyperinsulinemia, fasting hyperglycemia, failed reaction on insulin. Most affected patients survive up to 15 years.
	Autosomal dominant or recessive mutations	Type A insulin resistance syndrome (TAIRS)	Syndrome has a relatively good prognosis. Affected patients can live beyond middle age. Hypoglycemia is observed.
	Autosomal dominant mutations	Type C insulin resistance	It is a variant of Type A insulin resistance with less severe insulin resistance. It is so-called HAIR-AN (hyperandrogenic, insulin resistance, acanthosis nigricans).
	An autoimmune disorder caused by polyclonal autoantibodies acting against the insulin receptor.	Type B insulin resistance (TBIRS)	Depending on levels of autoantibodies, hypoglycemia or hyperglycemia may be observed.
	Depending on the type of lipodystrophy, these syndromes may be due to autosomal dominant, autosomal recessive or X-linked mutations.	Lipodystrophies	The lipodystrophy syndromes are a heterogenous group of disorders. Depending on the disorder, different pathologies, such as severe insulin resistance, severe hypertriglyceridemia, pancreatitis, complete or partial loss of adipose tissue and depletion of lipid storage capacity, are observed.

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
