# Peer review of "Changes in Cells Associated with Insulin Resistance"

_ijms, 2024, doi:10.3390/ijms25042397_

Round 1

Reviewer 1 Report

Comments and Suggestions for Authors

The article "Changes in cells associated with insulin resistance" presents a very current and advanced research. The article should be published after some minor corrections, such as:

  1. Figures should be improved.
  1. A graphical figure is needed to show the significance of hyperinsulinemia in various diseases.
  2. Clinical data linked with hyperinsulinemia should be provided in a tabular form.
  3. It is advised for authors to briefly include possible strategies against insulin resistance.
  4. The significance of this review should be provided.
  5. The strategies to characterize insulin in vivo should be described.
  6. Clinical studies linked to insulin resistance should be provided.
  7. The limitations of this study were not provided.
  8. Improve the writing of the objectives of this study.
  9. Revise the concluding section in light of this excellent evaluation.
  10. The English grammar and syntax of the manuscript need to be improved.

Comments on the Quality of English Language
  1. The English grammar and syntax of the manuscript need to be improved.

Author Response

Dear Reviewer

Reviewer 2 Report

Comments and Suggestions for Authors

General

It is a very extensive review on insulin resistance. However, it largely lacks references and is in need of English editing. To make it easier to envision the summarized information I would also suggest adding more figure, for example signaling pathways, insulin’s actions on different organs and/or an image demonstrating how insulin resistance arises. Although not the main scope of the review I would add a section on diabetes 1 and 2, especially as insulin resistance and T2D goes somewhat hand in hand. I also feel that the review should point out what is not yet known yet and needs to be parsed out in future studies.

Minor

Make sure that you’re using abbreviations throughout and have written out the abbreviations in full the first time that they are used. For example type 2 diabetes mellitus is written out after T2DM has been given as an abbreviation.

I think it would be good to add a discussion on sex differences in insulin resistance.

Here are some more specific comments:

Abstract

For the abstract I would make it a bit more general rather than a list of all of insulin’s actions. Maybe touch upon what systems it acts on in general, then what happens if things go wrong and then the purpose of the review in greater detail or what is discussed.

Here are some suggestions to improve the text if some of these sections are still included. For example, in the last sentence you say the review is mostly about cellular changes during insulin resistance. I think the abstract should better reflect that.

·         Add “a”. It plays an important role as a metabolic hormone.

·         Would suggest to reword: Insulin influences glucose metabolism..

·         It is involved in fat metabolism, by increasing storage of triglycerides in fast cells and decreasing lipolysis.

·         Insulin also reduces ketone body concentrations.

·         I feel like this sentence is repeating the one prior -  that sensitivity for insulin is reduced (I would delete it): It exists when glucose disposal in response to a particular concentration of insulin is impaired.

Introduction

·         You are just adding one citation at the end of each section. You need to add a reference for each statement please.

·         Insulin is a hormone that regulates different processes in the human body.

·         Is this the correct place to mention brain? Insulin regulated energy storage and metabolism partly through actions in the brain, but energy itself is stored in peripheral tissues? “Insulin regulates of energy storage and metabolism in organs and tissues such as the liver, kidney, brain, adipose tissue and skeletal muscle.”

·         In peripheral muscle, insulin stimulates metabolism, increases glucose uptake, glycogen synthesis and increases of muscle mass.

·         In the brain, this hormone is involved in stimulation of processes associated with hunger.

·         All of this is in adipocytes, correct? Because when you say “On the other hand” in the second sentence it sounds like we’re comparing between two different things. See suggested changes below: In adipose tissue, insulin stimulates processes such as metabolism and uptake of glucose, fat storage in lipogenesis and fatty acid transportation from the blood stream into cells. Insulin also inhibits lipolysis.

·         Do you mean perhaps mediated or regulated instead of associated?: Above mentioned processes are associated with intracellular signaling pathway [1].

·         This pathophysiological reaction increases plasma insulin levels, resulting in hyperinsulinemia which is associated with and increased risk of initiation, progression and metastasis of several cancers (ref). Hyperinsulinemia has also been linked to poor cancer outcomes, as well as development of metabolic diseases, such as type 2 diabetes mellitus (T2DM), metabolic  syndrome (ref). I would say that the latter is probably more well-known so I would mention insulin’s role in T2D and metabolic disease first and cancer second.

·         In developing of insulin resistance are involved genetic and environmental factors which are due to mutations in genes associated with intracellular insulin signaling pathway. Rewrite: Genetic and environmental factors contribute to the development of insulin resistance (ref). Specifically, in mutations in genes that are associated with the intracellular insulin signaling pathway (ref). These mutations can alter the structure of the insulin protein and/or the insulin receptor and can also affect insulin signaling (refs). Other mutations can affect insulin metabolism [2].

Characterization of insulin

·         This whole section lacks references: Insulin controls a wide variety of biological processes. Activation of the insulin receptor (INSR) by hormone binding, initiates signaling pathways that lead to the activation of enzymes which control many aspects of metabolism and growth which will be further described below?. Perturbations in these signaling pathways may cause for example insulin resistance, hypertension and/or T2DM.

·         It has previously has been suggested that insulin is solely produced by pancreatic β-cells located in Langerhans islets, however, recent observations have revealed that low concentrations of insulin are also produced in certain neurons of the central nervous system [3].

·         Insulin’s biosynthesis and secretion are controlled by circulating glucose levels. Explain – they’re increased when glucose levels are high.

·         I would delete this sentence, because it isn’t adding to the previous sentence and you explain in more detail in the sentence after: These two processes are inhibited by specific concentration of glucose.

·         The biosynthesis of insulin is stimulated when the concentration of glucose is between 2 mM and 4 mM and secretion is inhibited when glucose levels rise above 5 mM [4].

·         Would merge and alter these sentences: After secretion, the hormone circulates systemically, reaching cells in the liver, muscle and fat, where it is taken up and stored, resulting in reduced levels of glucose in the blood [5].

Biosynthesis of insulin – After this section I will not fix language but I see that it still needs to be edited and references also need to be added. Many short sentences can be put together to make it easier to read.

·         In section 2.1.1, when you explain that this is how a process takes place in humans it gives the idea that you’re going to also explain it in another species. 

·         Figure 1 may be more appealing if you add pictures in some way, for example where in the cell each step takes place or the picture of the protein.

3.1 Insulin receptor

·         Typically, you write out the name or word in full and the abbreviation in parenthesis. There is also hybrid receptor. In this case, heterotetramer is composed of an α/β dimer of INSR and an α/β dimer of insulin-like growth factor-1 receptor (IGF-1R). Hybrid receptor binds preferentially to insulin-like growth factor-1 (IGF-1), insulin-like growth factor-2 (IGF-2) over insulin [18].

·         I would rewrite the sentence as well: Insulin-like growth factor-1 (IGF-1) and insulin-like growth factor-2 (IGF-2) bind with much higher affinity to the hybrid receptor than insulin [18]. If you want you can also specify differences in affinity. You might also want to mention that IGF can bind to insulin receptors as well, considering that this section is about the insulin receptor.

·         In the figure legend for figure 2, please elaborate what the figure shows.

3.2 Substrates of insulin receptor

·         Insulin binds to the α-subunit of INSR, inducing a conformational change.

3.3

·         AKT is involved in many cellular functions.

3.4

·         Again, write out first then abbreviate in parenthesis: Activated INSR and IRS pro-161 teins possess docking sites for adaptor molecules which contain SH2 domains, such as 162 growth factor receptor-bound protein 2 (Grb2) and Shc (Src homology).

·         Please add references after each designated sentence and not all at the end of the paragraph, example this section: Then, activated ERKs are translocated to the nucleus, where they cause phosphorylation and transcriptional activation by transcriptional factors, such as ELK1, promoting cell division, protein synthesis and cell growth [1,20,23,27].

5.1

·         Hyperglycemia which may precede hyperglycemia? These patients may show hyperglycemia which may precede hyperglycemia [63,64].

5.2

·         I would not start a new section with “However,”.

·         5.2.1 Insulin resistance in select tissues and organs OR Insulin resistance in specific tissues and organs

·         This was an odd sentence, here are some suggested changes: Skeletal muscle is an important tissue involved in glucose metabolism in which glucose uptake is stimulated by insulin. Considering your next sentence maybe you could add that skeletal muscle is responsible for up to 80% of postprandial glucose uptake from the circulation or something similar.

·         I’m not sure what this means: ‘performed observations’?

·         The increased degradation of protein in insulin-resistant muscle due to defects in insulin cell signaling was confirmed also in other study (REF).

·         5.2.2.1 I don’t understand why the section starts with “According to this suggestion” as it’s a new section and you’re not really suggesting anything specific in the sentence prior.

·         5.2.2.2 Don’t understand this sentence either: The Randle cycle does not satisfactorily explain insulin resistance induced by lipid. Do you mean induced by excessive lipid accumulation for example?

Comments on the Quality of English Language

The English in this review needs to be edited. 

Author Response

Dear Reviewer

Round 2

Reviewer 2 Report

Comments and Suggestions for Authors

The author has made some of the suggestions from my review of this manuscript but has not, in my opinion, made all of the necessary changes. Many of the sections still have only one reference although they are providing a lot of information. I understand that some of the information is taken from one review, but you should still try to reference the original papers as much as possible.

The manuscript still needs some editing for the English used. If you do not want to send out (and pay for editing) may I suggest that you run the text through, for example, ChatGPT. You can add your text in and ask it to correct your grammar. You can also ask it to list all of the changes that it has made in the text after. This is just a tip and it's free to use.

Pertaining to the sections that I asked you to add. I understand that many topics are broad and can require full chapters to cover all of the information. I still feel as they would be relevant to include, and you could in the least add a paragraph and then suggest that the reader continue reading on the topic by citing a few reviews if they are interested in learning more.

Comments on the Quality of English Language

Although I asked them to edit the text in my previous correspondence, I do not feel as it has been adequately edited for publication. 

Author Response

Dear Reviewer

Again thank you very much for your opinion and suggestions.

According to your suggestions:

  1. Manuscript was corrected by professional proofreader, who has an office for proofreading.
  2. In many sections were added new references. In text there are added more than 60 new references. Therefore, there are more than one reference, however still are few information only with one reference. But this one reference is the most important. Most of these references are original articles.
  3. I have added one paragraph (7). I think that it will suggest for readers continue reading.

I think that after changes, article, if will be accepted for publication, will be more easy and friendly for readers.

Round 3

Reviewer 2 Report

Comments and Suggestions for Authors

The manuscript is much easier to read in its current state and I enjoyed reading the new sections that were added. I have no more comments.